# Nociceptor–Macrophage Interactions in Apical Periodontitis: How Biomolecules Link Inflammation with Pain

**DOI:** 10.3390/biom13081193

**Published:** 2023-07-31

**Authors:** Nandita Menon, Anil Kishen

**Affiliations:** Dental Research Institute, Faculty of Dentistry, University of Toronto, Toronto, ON M5G 1G6, Canada; nandita.menon@mail.utoronto.ca

**Keywords:** nociceptors, macrophages, neuropeptides, substance P, Calcitonin Gene-Related Peptide, pain, neurogenic inflammation, apical periodontitis

## Abstract

Periradicular tissues have a rich supply of peripheral afferent neurons, also known as nociceptive neurons, originating from the trigeminal nerve. While their primary function is to relay pain signals to the brain, these are known to be involved in modulating innate and adaptive immunity by initiating neurogenic inflammation (NI). Studies have investigated neuroanatomy and measured the levels of biomolecules such as cytokines and neuropeptides in human saliva, gingival crevicular fluid, or blood/serum samples in apical periodontitis (AP) to validate the possible role of trigeminal nociceptors in inflammation and tissue regeneration. However, the contributions of nociceptors and the mechanisms involved in the neuro-immune interactions in AP are not fully understood. This narrative review addresses the complex biomolecular interactions of trigeminal nociceptors with macrophages, the effector cells of the innate immune system, in the clinical manifestations of AP.

## 1. Introduction

Apical periodontitis (AP) is an inflammatory disease of periradicular tissue caused by oral opportunistic bacteria and their toxins retained in the apical regions of the periodontium [1,2]. Rich in blood vessels and nerve fibers, the periradicular tissue is composed of fibroblasts, epithelial cells, odontoblasts, cementoblasts, osteoblasts, osteoclasts, macrophages, and undifferentiated stem cells that form the periodontal ligament (PDL) and alveolar bone [3,4]. An infected root canal system in the apical periodontium causes all these immune cells and precursor cells to interact with one another, thereby eliciting an immune response against pathogens and pathogen-associated molecular patterns (PAMPs) [5]. However, AP can persist asymptomatically, without common clinical symptoms such as pain and swelling, while afferent sensory neurons that typically relay pain continue to engage in immune responses at the molecular level [6,7].

Neurogenic inflammation (NI) is set in motion when neuropeptides such as Substance P (SP) and Calcitonin Gene-Related Peptide (CGRP), among others, released by activated nociceptors act on resident immune cells and endothelial cells, causing vascular changes resulting in an influx of migrating immune cells to an inflamed tissue. To facilitate their function in NI, endothelial cells (vascular cells), mast cells, dendritic cells, T and B lymphocytes, and macrophages express cell-surface receptors for neuropeptides released by nociceptors [8,9]. Neuropeptides activate immune cells which release proinflammatory cytokines such as tumor necrosis factor (TNF)-α, interleukin (IL)-1β, IL-6, nerve growth factor (NGF), and prostaglandin E2 (PGE2) that are in turn known to activate sensory neurons [10,11]. Neuropeptides also play a role in regulating anti-inflammatory roles of immune cells. For example, CGRP has been shown to inhibit the production of pro-inflammatory cytokines and enhance the expression of a regulatory phenotype in activated macrophages [12]. This reflects on the pleiotropic nature of neuropeptides that are capable of modulating immune response. 

It is important to note that neurogenic inflammation is caused by the activation of peripheral nociceptors, resulting in inflammatory events in tissues that involve host innate and adaptive immune systems. Neuroinflammation, however, describes a localized host immune response to inflammation specifically in the brain and spinal cord [13,14].

Of the myriad cellular interactions in periradicular tissues, the cross-talk between macrophages and sensory neurons stands out because macrophages constitute up to 46% of inflammatory cells in AP [7]. This review aims to provide a comprehensive understanding of the temporal interactions between sensory neurons and macrophages while highlighting the role of nociceptors in different clinical manifestations of chronic AP. 

## 2. Overview of Macrophages in Apical Periodontitis

Inflammation is the physiological response to injury, stress, or infection. In terms of the immune cells involved, inflammation is the state in which innate and adaptive cells are activated, undergo differentiation, and migrate to resolve inflammation while restoring tissue homeostasis. A rapid host immune response is capable of resolving acute inflammation. On the other hand, prolonged exposure to triggers such as damage-associated molecular patterns (DAMPs), PAMPS, and pathogens results in a state of chronic inflammation that sees different immune and non-immune cells in action [15].

Acute AP is marked by the presence of periapical abscesses, while periapical granulomas are typically a sign of chronic AP, typically diagnosed via the radiographic visualization of rarefying osteitis in periapical tissues. AP is dynamic in nature because of the different immune cell types participating in different stages of AP. This dynamic nature can be observed via the progressive change in the distribution of immune cells in the periapical inflammatory infiltrate [16,17]. Immune cells can also undergo polarization to different phenotypes in order to facilitate the host immune response in disease or wound-healing [18,19]. 

Physiologically, the PDL has a key role in responding to microbes located within the pulp space. The PDL is a highly vascularized tissue, with up to ten times higher vascular volume than other fibrous connective tissues [20]. The blood vessels in PDL primarily participate in tooth support and shock absorption [21]. The activation of immune cells such as macrophages leads to the release of proinflammatory cytokines that increase the expression of cellular adhesion molecule (CAM-1) in endothelial cells, thereby promoting the local adhesion of monocytes and leukocytes [18,22]. This local response, combined with vasodilation caused by initial inflammation response, leads to the active infiltration of monocytes to the site of inflammation, where growth factors, pro-inflammatory cytokines, and microbial products induce cellular differentiation [23,24]. Once at the site of inflammation, these circulatory monocytes can be differentiated into macrophages or dendritic cells [25].

Toll-like receptor-4 (TLR4) is a sensory receptor predominantly expressed by human myeloid cells (monocytes and granulocytes) that recognize PAMPs such as lipopolysaccharides (LPS) expressed by Gram-negative bacteria [26]. Activated TLR4 recruits adaptor proteins that undergo phosphorylation. Myeloid differentiation primary response 88 (MyD88), Interleukin-1 receptor-associated kinase (IRAK), and tumor necrosis factor receptor-associated factor-6 (TRAF-6) are examples of such adaptor proteins. Phosphorylated adaptor proteins activate Nuclear Factor kappa B (NFκB) and mitogen-activated protein kinase (MAPK) signaling pathways, resulting in the expression of pro-inflammatory cytokines [27]. Accessory molecules such as myeloid differentiation 2 (MD2), cluster of differentiation (CD)36, and CD14 can expedite TLR4 signaling [28]. Macrophages are also antigen-presenting cells that activate members of the adaptive immunity, effectively mounting an adaptive immune response.

Macrophages are known to play an active role in resolving inflammation [29]. Activated macrophages can polarize into functional phenotypes of classically activated M1, or alternatively activated M2, to bring about pro- and anti-inflammatory effects, respectively. M1 macrophages produce pro-inflammatory cytokines TNF-α, IL-1β, and IL-6, which stimulate the production of tissue-damaging proteases and prostaglandins while mediating bone resorption. M1 macrophages also produce chemokines such as C-X-C motif chemokine ligand (CXCL)9 and CXCL10 that attract members of the adaptive immunity to the site of inflammation [30,31]. M2 macrophages are known to produce anti-inflammatory cytokines Transforming Growth Factor (TGF)-β and IL-10, which downregulate inflammation and immune response [32,33]. Their phagocytic capabilities help clear apoptotic cells and debris in healing tissues [5,34].

## 3. Role of Macrophages in Alveolar Bone Resorption

Resident macrophages in periodontal tissues secrete inflammatory cytokines and chemokines in response to pathogens. Qualitative and quantitative analyses have identified macrophages as the major constituent immune cell type in periapical granulomas [35]. This can be partly attributed to the vasodilation caused by the initial immune response, which in turn leads to the active infiltration of monocytes to the site of inflammation, wherein the growth factors, pro-inflammatory cytokines, and microbial products would induce differentiation to macrophages [24,36].

Alveolar bone resorption (loss) is a notable radiographic indication of chronic AP. The alveolar bone functions to hold teeth in the socket while absorbing any force applied to the teeth during biting and chewing. Subject to varying degrees of mechanical forces, the alveolar bone undergoes repeating cycles of bone formation and resorption. In a state of proinflammation, this equilibrium tends towards bone resorption, resulting in the loss of both mineral and organic components of the alveolar bone. The close association of macrophages with different stages of AP implicates these cells in the development of alveolar bone resorption. Macrophages were observed near osteoblasts in rat models of AP during the initial stages of bone resorption, while no macrophages were seen near the surfaces undergoing bone formation [19]. However, a study comparing data from immunohistochemical analyses of human specimens with clinical and radiological data reported no significant difference in the presence of macrophages between periapical granulomas and cysts [16]. Previous studies have highlighted disparities in the clinical presentation of AP [16,37]. Thus, a better understanding of the mechanisms underlying bone modeling and remodeling will help in identifying molecular pathways linking macrophages to alveolar bone resorption.

Osteoclastogenesis, or the development of mature osteoclasts (bone resorptive cells) from precursor cells, is mediated by receptor activator of nuclear factor kappa-β ligand (RANKL) [38]. RANKL is a cytokine produced by osteoblasts (which synthesize bone matrix and differentiate into osteocytes [39]) and stromal cells that binds to its receptor, RANK, to initiate the differentiation of osteoclast precursor cells [40]. This process is regulated by Osteoprotegerin (OPG), a decoy receptor of RANKL that is primarily produced by osteoblasts [41]. Hence, the RANKL-RANK-OPG trio are the main players in bone modeling and remodeling pathways [42]. GM-CSF (Granulocyte-Macrophage Colony-Stimulating Factor) is a cytokine that induces proliferation and maturation in macrophages, and it has been found to promote fusion of prefusion osteoclasts in the presence of RANKL, enabling bone resorption under pathologic and inflammatory conditions [43,44]. Besides GM-CSF, macrophages produce an array of inflammatory mediators in response to pathogens or microbial toxins that have been linked to bone resorption in AP [45]. IL-1 is one such proinflammatory cytokine that is produced by activated macrophages, which promote bone loss by enhancing RANKL expression by stimulating the synthesis of PGE2 in osteoblasts [46,47]. In addition, IL-1 is involved in the survival of osteoclasts and is known to induce multinucleation of osteoclasts, which is a hallmark of cell maturation [48]. Macrophages produce TNF-α, a proinflammatory cytokine that inhibits the differentiation and production of matrix proteins in osteoblasts [49,50]. TNF-α was found to play a role in osteoclastogenesis when increased RANKL mRNA expression was observed in osteocytes (which modulate bone remodeling [45]) that were cultured in vitro with exogenous TNF-α. This study observed an increased percentage of RANKL-positive osteocytes in mice injected with TNF-α [51].

## 4. Sensory Neuronal Activation in Apical Periodontitis

Periradicular tissues have an abundant supply of neurons that originate in the trigeminal ganglion (TG) [52]. Research groups in the 1970s identified nociceptive afferents with different activation thresholds and conduction velocities in the PDL [53]. Immunocytochemical analysis identified NGF-positive sensory nerves in periodontal tissues from rats [54], while ultrastructural analyses of the human PDL using electron microscopy identified Ruffini nerve endings, free nerve endings, and lamellated corpuscles [55]. Electrophysiological studies performed on cat models helped characterize trigeminal nerve endings in the PDL into low-threshold mechanoreceptors sensitive to mechanical load and vibration (A-β fibers) and nociceptors sensitive to pressure, temperature, and chemical irritants (A-δ, C-fibers) [56,57]. A-β fibers have large cell bodies, and these are myelinated [58,59]. A-δ fibers have medium-sized cell bodies, and C-fibers are considered the smallest [58,59,60]. Myelinated A-δ fibers have higher conduction velocities, resulting in a sharp pain sensation compared to the unmyelinated, slow C-fibers, which relay a delayed and dull, ache-like pain [58,61]. 

Studies implicating sensory neurons in inflammation have come a long way since the classic studies that concluded that neurogenic inflammation is based on axon reflexes [62,63]. While this theory did not pan out, some of the earlier works were successfully able to link activated sensory neurons to inflammatory phenomena such as vasodilation, setting stage for further studies [64,65]. As with the classical inflammatory response, neurogenic inflammation has also been associated with tissue regeneration [66]. It was reported that sensory denervation in rats, with pulp exposure to oral environment, showed the loss of pulpal tissue, while innervated teeth showed extensive neural sprouting and formation of reparative dentin and osteodentin [66].

Specialized surface receptors expressed by nociceptors play a crucial role in the cascade of events that led to NI (Figure 1). Transient receptor potential (TRP) channels are a class of integral membrane proteins that function as signal transducers in sensory neurons. These are polymodal in nature and can be activated by intracellular and/or extracellular mechanical, thermal, chemical, and osmotic pressure gradients. Based on sequence similarity, several classes of TRP channels exist. The TRPV subfamily (vanilloid subtype) has six members, of which TRPV1 has garnered attention for its role in the detection of noxious physical and chemical stimuli [67,68]. TRPA1 is the sole member of the TRPA subfamily (ankyrin subtype) [69,70]. The detailed 3D structure of the TRPA1 receptor was previously characterized by Cao et al. and Liao et al. [71,72]. Such studies made it possible to study the conformational changes that occur when TRPA1/TRPV1 receptors are activated. Their activation results in an active influx of extracellular calcium ions, which mediate action potentials in neurons and trigger exocytosis of CGRP and SP from large dense core vesicles (LDCVs) that involve members of the SNARE (soluble N-ethylmaleimide-sensitive factor attachment protein receptor) family of proteins [73,74]. Biologically active neuropeptides are derived from their precursor peptides that are cleaved in the endoplasmic reticulum, after which these pro-peptides undergo post-translational modifications to form active versions that are stored in LDCVs prior to release from the sensory nerve endings [75,76]. A-δ and C fibers express TRPV1 and TRPA1 channels, and hence, these channels are associated with chemical and thermal nociception [77]. Knockout models of TRPV1 and TRPA1 highlight the role of these receptors in eliciting host immune response besides nociception. Non-neuronal cells such as endothelial cells, fibroblast, odontoblasts, and immune cells, including mast cells, monocytes, macrophages, and dendritic cells, also express TRPV1 and TRPA1 [77,78,79]. In these cells too, TRPV1 and TRPA1 exhibit functional responses to similar triggers as seen in neurons; however, it is not clear yet if the sensitivity thresholds of these receptors in non-neuronal cells are similar to those in neurons [77,80].

Toll-like receptors (TLRs) are another class of surface receptors that have been identified on nociceptors. The nociceptor expression of TLR4 highlights its role in the host immune response to bacterial infections [81,82]. Peripheral sensory neurons showed the colocalized expression of TLR4 and CD14, indicating the capability of the direct detection of PAMPs [82]. Besides immunological implications, the LPS binding of TLR4 on trigeminal neurons was shown to increase sensitization of TRPV1 channels, leading to increased expression of CGRP, which suggests a possible mechanism for pain associated with bacterial infections [83,84,85]. Similar findings were reported in an independent study that demonstrated a concentration-dependent effect of LPS on TRPV1 in trigeminal neurons [83].

Different bacterial strains have been shown to evoke different degrees of nociceptor response [86]. Gram-positive bacteria operate through mechanisms that are distinct from those triggered by Gram-negative bacteria [87]. *S. aureus* utilize N-formylated peptides and pore-forming toxin α-hemolysin to instigate pain in mice [87]. This study also showed a strong correlation between the bacterial load of live *S. aureus* and pain expressed by mice. Such studies provide insight into the different mechanisms that may be involved in eliciting pain in bacteria-mediated diseases such as AP, where both Gram-negative and Gram-positive strains play active roles in disease pathogenesis [2,88].

## 5. Role of Sensory Neuropeptides in Pain and Inflammation

Neuropeptides are considered the primary mediators of neurogenic inflammation. A-δ and C-fibers are peptidergic in nature, meaning these release neuropeptides mediating NI in AP [89,90] (Table 1). CGRP and SP are the principal neuropeptides involved in AP.

CGRP was discovered in 1982 by Amara et al. [135]. This 37-residue peptide was found to be expressed in the central and peripheral nervous systems, promoting sensory, motor, and autonomic functions in the brain [136]. CGRP has two isoforms, α-CGRP and β-CGRP, which have over 90% similar homology and share similar biological effects in humans [75]. CALC I gene undergoes alternate splicing to generate α-CGRP, while β-CGRP is a product of CALC II gene transcription. While α-CGRP is predominant in the central and peripheral nervous systems, β-CGRP is found mainly in the enteric nervous system [75]. Vasodilatory effects of human and rat CGRP were discovered in 1985, effectively linking CGRP to hyperemia [137]. The development of CGRP antagonists helped understand the role of CGRP in migraines, linking neuropeptides with pain pathways in the orofacial region and in other tissues [138]. Recognizing the ability of sensory nerves to produce and release CGRP led to the discovery of different stimuli that can activate neural cells to release this neuropeptide and the pathways that lead to the production and release post-activation [139].

The CGRP receptor molecule is a G protein-coupled receptor and is composed of three components: receptor activity modifying protein 1 (RAMP1), Calcitonin-like receptor (CLR), and receptor component protein (RCP). All three components are required to form a fully functional CGRP receptor (CGRP-R). Upon activation by binding with CGRP, CGRP-R may couple with G protein α subunit, increasing the cyclical adenosine monophosphate (cAMP) levels, which activate protein kinase A (PKA). This can result in the phosphorylation of cAMP response-element-binding (CREB) protein leading to gene transcription or the activation of extracellular signal-regulated kinases (ERK) pathway within a neural cell [139]. This signaling pathway can also result in the opening of ATP-sensitive K^+^ channels, which can lead to vasodilation [98]. 

Substance P (SP) is an undecapeptide that was first discovered in 1931 by Von Euler and Gaddum [140]. SP, neurokinin A (NKA), neuropeptide K (NPK), and neuropeptide γ (NPγ) are tachykinin neuropeptides encoded by the TAC1 gene; alternate RNA splicing and differential post-translational processing result in the expression of specific tachykinin peptides [141]. SP is best known for its role as a modulator of pain perception by altering cellular signaling pathways using G protein-coupled receptors called neurokinin 1 receptor (NK-1R) that can act through the cAMP secondary messenger system or inositol triphosphate (IP3)-diacylglycerol (DAG) system [108]. These downstream signaling pathways result in the expression of pro-inflammatory cytokines. SP can also exhibit autocrine effects on neural cells expressing NK-1R in a desensitizing process that results in the internal phosphorylation of NK-1R and recycling via endocytosis and acidification in endosomes [108].

CGRP and SP are known to colocalize, and these have been studied to assess the degree of pain experienced by patients with symptomatic AP [142]. Affirming their role in oral diseases, several studies have reported increased levels of CGRP and SP in the sites of inflammation in the dental pulp [143,144], periodontal ligament [145,146], and saliva and gingival crevicular fluid [111,147,148]. The levels of neuropeptides in these environments are correlated with the degrees of inflammation, confirming their role in inflammatory mechanisms [111,113,144].

CGRP and SP play vital roles in modulating bone resorption. In murine AP models, TRPV1-positive sensory neurons were shown to regulate osteoclastogenesis via CGRP signaling [104]. On the other hand, SP was found to up-regulate RANKL/OPG ratio in a rat model of periodontitis, promoting RANKL-induced osteoclast differentiation [115].

SP and CGRP have also been shown to mediate tissue repair. The sprouting of CGRP-positive fibers in odontoblast layers after injury induced in the cervical dentin of rats resulted in the formation of reparative dentin [149]. SP released by stimulated peripheral nerves increased blood flow to the dental pulp, thereby enabling tissue healing [113,116]. Hence, SP and CGRP can be considered immunomodulators that can have pro- and anti-inflammatory effects depending on molecular cues present in the tissue microenvironment.

## 6. Neuro-Immune Interactions Modulate Inflammation and Healing in AP

Evidence of a functional link between sensory neurons and immune cells comes from the proximity of sensory nerve endings and immune cells. This is corroborated by the expression of neuropeptide-specific receptors on immune cells and cytokine receptors and receptors capable of recognizing microbial pathogens on trigeminal nociceptor neurons [8,150] (Figure 1 and Figure 2).

In vitro studies have confirmed the expression of CGRP-R and NK-1R in murine and human macrophages, implying that neuropeptides CGRP and SP influence the activities of these immune cells [99,151]. Pro-inflammatory cytokines IL-1, IL-4, and interferon (IFN)-γ were also shown to induce NK-1R expression in macrophages [152,153]. Goldman et al. were among the first to conduct in vitro studies on peritoneal macrophages, showing enhanced phagocytosis when exogenous SP was applied to these cells [154]. The SP-mediated activation of macrophages was discovered when exposure to SP increased levels of eicosanoids in macrophages [155]. SP also acts as a chemoattractant for monocytes and macrophages, first demonstrated in vitro using human monocytes and peritoneal macrophages from guinea pigs [156,157]. Evidence of TNF-α, IL-1, and IL-6 release in macrophages in response to SP was also previously recorded [158]. In contrast, CGRP was found to actively prevent the activation of monocytes and macrophages [102,103]. Exogenous CGRP reduced proliferation in peripheral blood mononuclear cells and inhibited macrophage production of hydrogen peroxide, which is considered a classic response to inflammation [102,103]. 

Early identification of CGRP immunoreactive (CGRP-IR) nerve fibers in the PDL led to increased interest in studying their response to different stimuli. Dynamic changes were observed in CGRP-IR nerve densities surrounding blood vessels in PDL during tooth movement in rat molars [159]. Wakisaka et al. were among the first to study the distribution of SP-positive nerve fibers in rat PDL [160]. However, when immunoreactions for CGRP, SP, and neuropeptide Y (NPY) were evaluated in dental pulp, periodontal ligament, and gingiva in cats, denser innervation in the pulp was reported, and these nerves were found to be more immunoreactive to CGRP than to SP [161]. A similar trend was observed by the same group in a ferret model, affirming a similar abundance in sensory nerve innervation across different species [162]. Hence, several subsequent studies focused on studying CGRP-positive neurons. Kimberly and Byers induced pulpitis that led to periapical lesions in rat models and monitored changes in CGRP-IR nerves to inflammation for 35 days [163]. They observed that while inflammation initially caused nerve damage, increased neural sprouting was seen along the wound border. Even as inflammation advanced in severity and transitioned from the pulp to the periapical regions, CGRP-IR nerves sprouted extensively in both tissues. Independent studies reported similar patterns of neural sprouting in rat models, linking the response of pulpal innervation to inflammation and the induction of periapical lesions [164].

Dynamic changes in the distribution of sensory nerves during acute and chronic AP have drawn attention to understanding how these specialized neurons may modulate disease progression. Several studies measured increased levels of neuropeptides in saliva and gingival crevicular fluid (GCF) in animal models and in patients diagnosed with AP [165,166]. Similarly, investigations on the distribution patterns of neuropeptides in periodontitis suggested the potential of using neuropeptides as biomarkers for the early detection of periodontitis [167,168]. A significant increase in SP-like immunoreactivity was measured in GCF collected from periodontitis sites of volunteers [169]. Studies also found a significant correlation between SP and proinflammatory IL-1β, TNF-α, IL-8, and chemoattractant MCP-1 (monocyte chemoattractant protein-1) in GCF from patients with periodontitis [170]. CGRP exhibited a different profile in comparison; reduced CGRP immunoreactivity was observed in periodontitis sites compared to healthy periodontium [97]. Further analysis showed that carboxypeptidase activity in GCF resulted in a rapid breakdown of CGRP; however, this did not affect the levels of SP in GCF [171].

Macrophages express functional CGRP-R, i.e., receptors of CGRP. CGRP activation led to the proliferation and differentiation of murine macrophages into osteoclast-like cells via a cAMP/PKA-dependent signaling pathway [172]. Murine bone-marrow-derived macrophages (BMDMs) showed increased IL-6 expression as a response to CGRP activation, which led to speculations about the role of BMDMs in the negative inhibition of B cell development [173]. Similar findings were reported in rat models that demonstrated reduced IL-6 expression in invading macrophages upon perineural injection of a CGRP antagonist [100]. In vitro studies have shown that LPS-induced expression of pro-inflammatory cytokines, including IL-1β, IL-6, contributes to CGRP expression in macrophages [99].

Primarily produced by sensory neurons in the periodontal tissues, NGF is a neurotrophic factor that plays a critical role in neurogenesis, promoting the growth, differentiation, and survival of neurons [174,175]. Nerve Growth Factor-Tropomyosin receptor kinase A (NGF/TrkA) signaling suppresses CGRP production in monocytes as well as RAW264.7 macrophages [99]. In periodontal tissues that have a rich sensory nerve supply, NGF can induce increased neural sprouting and innervation under inflammation. NGF also controls the expression and release of neuropeptides in sensory neurons. NGF-mediated neural responses have direct implications on the host tissue response to inflammation [176,177]. NGF has been shown to induce monocyte differentiation in macrophages, thereby enhancing inflammatory effects [178]. Non-neural cells such as fibroblasts and osteoblasts can produce NGF in periodontal tissues, and several non-neural cell types express TrkA, which is a high-affinity NGF receptor molecule [179,180]. Hematopoietic stem cells express TrkA and produce their own NGF to support differentiation in these cells. This expression is maintained so that neutrophils, mast cells, and monocytes that arise from these stem cells also express TrkA. An interesting finding was that the expression of TrkA in monocytes and macrophages had functional implications that increased when these cells were stimulated with bacterial toxins [178,181]. For example, TrkA-stimulated human monocytes showed an upregulation of anti-apoptotic genes [182]. Another study showed that monocytes production of CGRP was reduced in response to LPS when NGF was neutralized [183]. TrkA-dependent ruffling of human-monocyte-derived macrophage cell membrane, induced by NGF, was found to increase phagocytic activity [184]. NGF also induced increased the macrophage expression of matrix metalloproteinase (MMP)-9 in this study. Although the functions and release mechanisms of neurotrophins in immune cells remain elusive, current knowledge substantiates the role of sensory-neuron-derived NGF in macrophage functions under inflammation.

SP was found to bind NK-1R receptors on murine macrophages and activate protein kinase C (PKC) or phosphatidylinositol 3-kinase/protein kinase B (PI3K/Akt), which triggers nuclear factor kappa-light-chain-enhancer of activated B cells transactivation and chemokine response via ERK1/2 and p38 MAPK signaling pathways [185,186]. Both resulted in the expression of the chemokines macrophage inflammatory protein-2 (MIP-2) and MCP-1. MIP-2 is a potent chemoattractant for neutrophils, and MCP-1 regulates monocyte migration and infiltration [187,188]. Macrophages produce SP in response to NF-κB activation, linking this phenomenon with the pro-inflammatory M1 macrophage phenotype. The administration of exogenous SP has been found to increase IL-10 expression in rat models, creating conducive environment for the activation of anti-inflammatory M2 macrophages, which can also produce SP [189,190]. SP was found to induce hemeoxygenase-1 (HO-1), an antioxidant, in LPS-stimulated murine macrophages in a dose-dependent manner [191]. HO-1 is expressed in response to oxidative stress and functions to suppress inflammation by inducing macrophage polarization in favor of an M2 anti-inflammatory phenotype [192]. SP also induced M2-like macrophages by activating PI3K/Akt in murine macrophages [193]. Hence, there is evidence of both pro- and anti-inflammatory effects of SP, which warrants detailed studies in AP.

## 7. Association of Pain with Nociceptor-Macrophage Cross-talk in AP

Pain is a natural response to tissue damage, and current investigations highlight a close association of this sensation with nociceptor–immune-cell interactions [194]. Several aspects can contribute to the presentation of pain in endodontic infections, including the volume of root canal space, the presence of pathogen diversity, the virulence of pathogenic load, the state of the immune system in an individual, and lesion dynamics, among others [195]. Despite standardized systems such as Numeric Rating Scale (NRS-11) and periapical index (PAI), which can correlate pain with clinical presentations in AP, these cannot be used to predict when and why pain is not experienced by patients with chronic AP [196,197]. Understanding pain perception from the perspectives of periapical inflammation on a molecular level may help improve diagnosis and develop better treatment plans and strategies.

Immune cells modulate nociceptor activity using cytokines and chemokines, and nociceptors in turn modulate functions of immune cells by expressing neuropeptides, resulting in a dynamic, bi-directional cross-talk between these two cell entities during tissue injury and inflammation [198] (Figure 3). Nociceptors express receptor molecules that are activated by PAMPs, DAMPs, and mediators released by immune cells. Neutrophils, mast cells, and macrophages have close encounters with nociceptive nerve terminals, and hence, mediators produced by these immune cells can induce pain sensitization and influence chronicity of pain. Macrophages produce TNF-α, IL-1β, IL-6, PGE2, and Leukotriene B4 (LTB4), covering a range of cytokines, growth factors, and lipids that bind to specific receptors on nociceptor neurons and result in direct pain sensitization or engage ion channels to increase neuronal cell excitation which then causes pain sensitization [194]. Such pain sensitization translates to hyperalgesia or allodynia in AP.

Hyperalgesia and allodynia refer to increased nociceptive sensitivity in peripheral nerves. While both are caused by the reduced activation threshold of nociceptor neurons, hyperalgesia is caused by repeated or intense exposure to noxious mechanical or thermal stimuli, and allodynia is associated with non-toxic mechanical or thermal stimulation. A comparison of maximal bite force in contralateral, non-inflamed teeth in patients with acute periradicular periodontitis against teeth from healthy individuals reported allodynia in the contralateral teeth of acute AP patients. This was attributed to central sensitization, which refers to an increase in sensitivity in neurons in the central nervous system (CNS) [199,200]. Owatz et al. proposed that central sensitization may evoke mechanical allodynia in pulpal nociceptors independently of the local activation of the nociceptors in pulpal tissue [201]; this is yet to be confirmed. Studies have found correlations between nociceptive pain and neuropeptide CGRP [138,202]. Mechanical allodynia was reported when CGRP was injected into the hind paw skin of mice [203]. The intradermal injection of CGRP antagonist after 1 h of capsaicin injection resulted in a partially reversed hyperalgesia and mechanical allodynia in rats, confirming the role of CGRP in maintaining nociceptive sensitivity [204]. 

De França et al. evaluated M1 and M2 macrophage populations in human radicular cysts and periapical granulomas the presence of both are classic markers of apical periodontitis [205]. The authors also correlated pain with macrophage subpopulations in samples tested; a higher prevalence of macrophages was observed in areas with higher densities of microorganisms [205]. Based on sample collection, more cases with radicular cysts were symptomatic compared to periapical granulomas, meaning this trend could be attributed to the pro-inflammatory profile of cysts that were assumed to have increased levels of IL-1, IL-6, and PGE2, all of which are released by macrophages in response to microbial infection and have been identified in causing painful symptoms. However, periapical granulomas were characterized by pulsating pain in this study, and this was linked to higher inflammatory infiltrate that also had higher amounts of CD68^+^ cell populations. A correlation between macrophages and CGRP, known to initiate and maintain pain, was reported when the increased presence of CGRP-immunoreactive fibers was observed along with an increased number of resident macrophages in rat models of AP [206]. These studies have directly or indirectly highlighted an important link between macrophages and pain sensations in AP. There is, however, a lack of consensus that successfully links macrophages and pain in AP progression from radicular cysts to periapical granulomas [205].

A recent study found correlations between the immunoexpression of MMPs and the presence of pain in patients with AP; periapical granulomas and periapical cysts in teeth extracted from patients who were evaluated for the presence of pain showed high expression levels of MMP-2 [207]. MMPs are a class of enzymes that participate in the degradation of matrix proteins and bone, contributing to the processes of tissue degradation and remodeling. Categorized into different classes such as collagenases, gelatinases, stromelysins, and others, MMPs are also capable of cleaving cytokines and chemokines, thereby modulating immune response. MMP-1, produced by activated macrophages, has been found in increased concentrations in rat and mouse models of periapical lesions [208,209]. MMP-2, MMP-3, MMP-8, MMP-9, and MMP-13 have also been identified in human periapical lesions; of these, macrophages have been shown to express all of these in AP [210,211,212]. Although their mechanism of action is not fully understood, it is known that, in response to different cues, MMPs are secreted by different cell types into the extracellular matrix, where they are activated proteolytically and their actions are regulated by endogenous inhibitors called tissue inhibitors of metalloproteinases (TIMPs) [213,214]. In the context of AP, pro-inflammatory cytokines TNF-α and IL-1β are known to activate MMP-1 and MMP-2; macrophages are a major source of these cytokines [213,214]. Stimulating pain in the upper central incisor of human volunteers resulted in elevated levels of MMP-8 in the gingival crevicular fluid [112]. This study observed an increase in SP levels in the GCF of stimulated teeth, thereby correlating neuronal activation associated with pain with immune activation.

## 8. Conclusions

The discovery of neurogenic inflammation highlighted the role of nociceptors in the immune response. The dental pulp and periapical tissues are rich in sensory nerves, and nociceptors are now known to activate key elements of the innate immune system besides sending pain signals to indicate infection. Neuropeptides and inflammatory mediators released by both nociceptors and immune cells regulate neuro-immune cross-talk. Animal models and in vitro studies have led to a deeper understanding of neuro-immune cross-talk; however, we are yet to understand how microbial pathogens, immune cells, and nociceptors interact with each other resulting in the modulation of pain and inflammation in AP. Chronic AP is especially challenging to treat or manage because it is usually asymptomatic, persisting in a state of continued inflammation with varying presenting symptoms. Macrophages play a critical role in sensitizing and desensitizing nociceptors, leading to hyperalgesia, allodynia, and central sensitization in patients with AP. A comprehensive insight into the molecular mechanisms that connect neuro-immune interactions, inflammation, and pain is essential to understand periapical host response in symptomatic and asymptomatic AP while offering significant translational potential.

## Figures and Tables

**Figure 1 biomolecules-13-01193-f001:**
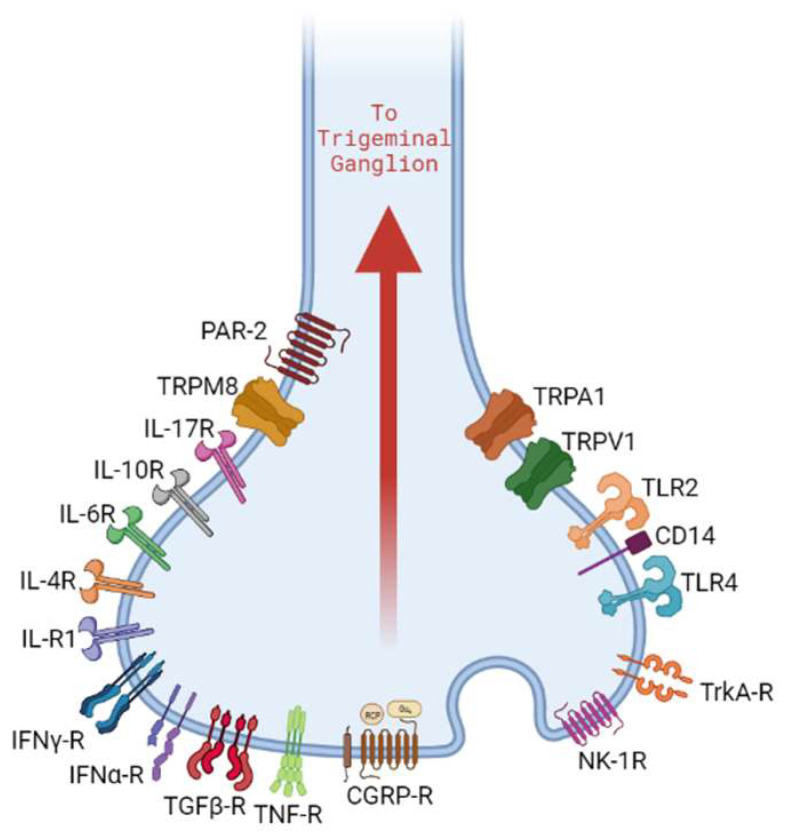
Nociceptor neurons express receptor molecules that recognize PAMPs, DAMPs, cytokines, and neuropeptides. This results in the activation of downstream signaling pathways that can contribute to neurogenic inflammation. Initialisms are listed in Appendix A.

**Figure 2 biomolecules-13-01193-f002:**
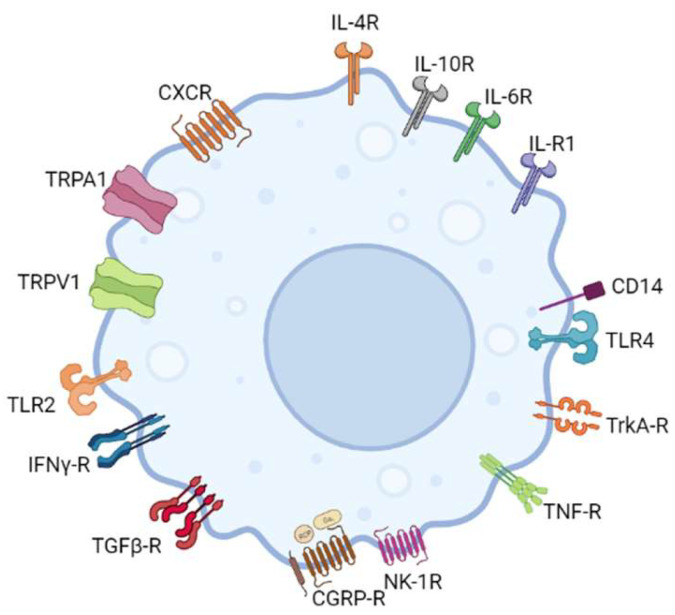
In addition to releasing neuropeptides, macrophages have been shown to express neuropeptide-specific receptors, indicating a significant role in neuro-immune cross-talk. Initialisms are listed in Appendix A.

**Figure 3 biomolecules-13-01193-f003:**
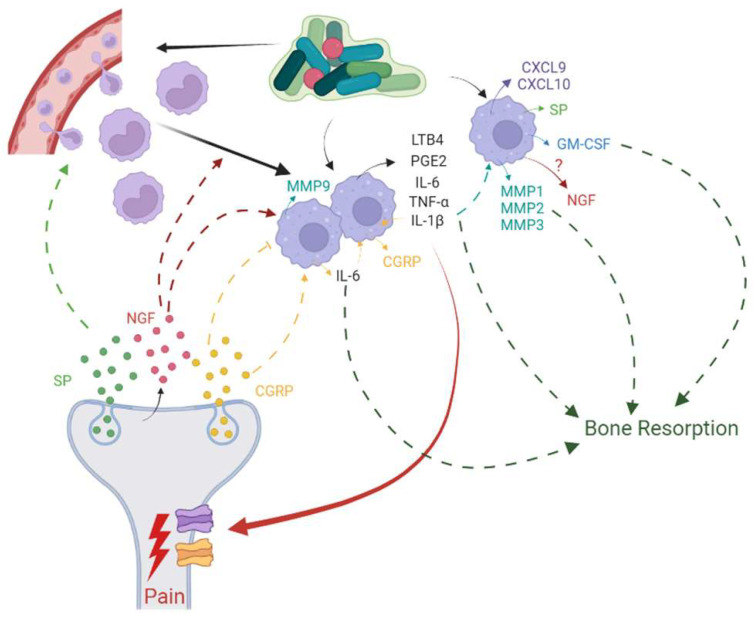
Bi-directional cross-talk between nociceptor neurons and macrophages has been implicated in inflammation, resulting in pain and bone resorption in apical periodontitis. Initialisms are listed in Appendix A.

**Table 1 biomolecules-13-01193-t001:** Neuropeptides (NPs) identified in apical periodontitis (AP), cells known to release NPs, cells shown to express receptors for NPs, and known functional effects of NPs in AP.

Neuropeptides in AP	Cells Releasing NPs	Cells Expressing NP Receptors	Functional Effects in AP
Calcitonin Gene-Related Peptide (CGRP)	Sensory neurons [9,75] T cells [91] Monocytes [91] Endothelial cells [75] Fibroblasts [92,93] Adipocytes [75]	Sensory neurons [90] Arterial vessels, Mononuclear immune cells, Schwann cells [91] Th1 cells [94] Langerhans cells [94] Monocytes [94] Macrophages [94] Dendritic cells [94] CD34^+^ hematopoietic progenitor cells [94] B & T lymphocytes [94] Epithelial cells [95]	Extensive degradation, resulting in reduced levels of CGRP observed in the GCF of diseased periodontal tissues [96,97] Vasodilation [98] Activation of innate and adaptive immune cells [99,100] Increase in blood flow [101] Prevention of monocyte/macrophage activation [102,103] Stimulation of osteogenesis [104,105] Endothelial cell proliferation [101]
Substance P (SP)	Neurons [106,107] Astrocytes [108] Microglia [107,108] Epithelial cells [108] Endothelial cells [108] T cells [108] Macrophages [108] Dendritic cells [106,108] Eosinophils [108] Mesenchymal stem cells [108] Leukocytes [106] Natural killer cells [106] Mast cells [106] Neutrophils [106]	Epithelial cells [108] Endothelial cells [106,108,109] White blood cells [109] Fibroblasts [108,109] Neurons [90,108,109] Smooth muscle cells [106,108] B & T lymphocytes [106,108] Natural killer cells [108] Dendritic cells [108,110] Monocytes [106,108] Macrophages [106,108] Microglia [106,108] Astrocytes [106,108] Eosinophils [106,108] Mast cells [106,108] Neutrophils [106]	Increased levels observed in GCF of diseased periodontal tissues [96,111] Desensitization of nociceptors [108] Neuronal activation [112] Onset of pain and inflammation [113] Inhibition of osteoblast differentiation [114] Induction of osteoclastogenesis [115] Increase in blood flow [113,116] Reduction in pro-inflammatory response of neutrophil, epithelial cells [117]
Vasoactive Intestinal Polypeptide (VIP)	Neurons [118] T cells [118] B cells [118] Mast cells [118] Eosinophils [118]	Epithelial cells [118] T cells [118] Macrophages [118] Dendritic cells [118] Mast cells [118] Neutrophils [118] Smooth muscle cells [118] Endothelial cells [118] Pancreatic B cells [118]	Found in increased levels in GCF of periodontitis patients; reduced levels as inflammation reduced [119] VIP reactivity observed in gingival biopsy, but no discernible difference compared to clinically healthy sites [120] Promotion of bone regeneration [105]
Neuropeptide Y (NPY)	Neurons [121] Pancreatic β cells [122] Vascular smooth muscle cells [123] Monocytes [123] Macrophages [123] Enterocytes [123] Endothelial cells [121] Langerhans cells [123] Microglia [123] B & T lymphocytes [123]	Neurons [121] Monocytes [123] Macrophages [122,123] Lymphocytes [123] Dendritic Cells [123] Neutrophils [123] B & T Lymphocytes [123] Leukocytes [123] Microglia [123] Osteoblasts [122] Endothelial Cells [122] PDL Stem Cells [105]	Proposed to be involved in initiation and regulation of periodontitis [105] Reduced levels observed in diseased sites of periodontitis patients [96] High levels observed in the saliva of periodontitis patients [124] Promotion of osteogenesis [105] Increase in PDL stem cells’ osteogenic capacity in vitro [125]
Pituitary Adenylate Cyclase-Activating Peptide (PACAP)	Mast cells [126] Schwann cells [127]	Pancreatic cells [128] Sensory nerve terminals [128,129] Pituitary cells [127] Schwann cells [127] Satellite glial cells [129] Hepatocytes [129] Osteoblasts [129]	Promotion of bone regeneration [105] Dose-dependent inhibition of IL-12 in LPS-activated macrophages [130]
Neurokinin A (NKA)	Sensory neurons [131]	Sensory neurons [132] B & T cells [133] Dendritic cells [133] Macrophages [133]	Co-released with SP [131] Increased levels observed in the GCF of diseased periodontal tissues [134] Decreased levels of NK-A observed in diseased periodontal tissue [96]

## Data Availability

No new data were created or analyzed in this study. Data sharing is not applicable to this article.

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
