# Peer review of "Nociceptor–Macrophage Interactions in Apical Periodontitis: How Biomolecules Link Inflammation with Pain"

_biomolecules, 2023, doi:10.3390/biom13081193_

Round 1

Reviewer 1 Report

This review article has been well written. It gives an in-depth coverage of the topic chosen. 

Reviewer 2 Report

Dear authors

First of all, I would like to thank you for the opportunity to collaborate in the review of the paper entitled Nociceptor-Macrophage Interactions in Apical Periodontitis:Biomolecules Linking Inflammation to Pain. This is a narrative review that forwards the complex biomolecular interactions of TG nociceptors with macrophages, effector cells of the innate immune system, in the progression  and resolution of AP.

In my opinion, the subject is relevant, however, how the work is presented hinders the reader's interest. I highlight some points that should be rethought initially

Title

Nociceptor-Macrophage Interactions in Apical Periodontitis: Biomolecules Linking Inflammation to Pain. The title should be more eye-catching. Maybe something tending towards the following suggestion: The interaction of PA with pain: how to understand this phenomenon?

Abstract

The objective must be clearer. I suggest that the entire abstract be redone

Considering the keywords, one should not abbreviate.

Introduction

In line 48 “Sensory nociceptors in AP” I believe that all nociceptors are sensory

In the introduction, the objective should be at the end. “Herein we aim to provide a comprehensive understanding of the temporal interactions of sensory neurons and  macrophages, correlating these with different clinical manifestations of AP thereby highlighting the role of sensory nociceptors in AP”

Was the authors' intention to differentiate between neurogenic inflammation and neuroinflammation? It wasn't clear (lines from 53 to 58)

Line 58-61 remove please” This review identifies with the definition of neurogenic inflammation used to describe inflammatory response of activated specialized nociceptors in the host tissues while aiming to address studies characterizing NI in chronic AP.“ It's confusing

 In this 2. Modulation of Inflammatory Response in Apical Periodontitis

AP is typically diagnosed by radiographic visualization of rarefying osteitis in periapical tissues. Acute or chronic?

Lines from 75 to 77  “Efforts to understand the role of specific immune cells in the progression of AP has generated conflicting data about the distribution of major cell types in the periapical inflammatory infiltrate[23]”. Do you mean it was not answered? Since this is a very old article.

Very old references mostly!!!!

When reading the work, I felt in the place of the readers. Despite being a relevant subject, it is written in a way that does not excite the reader. The work is extensive, with old literature (references).

From lines 107 to 113, what is the importance of this for the work? This is very confusing.

From lines 141 to 144, the authors were not clear in describing the PA dynamics.

 Conclusion

I noticed that the conclusion was too long for a paper published in the journal

I suggest summarizing the work and removing excess references.

In short

I suggest that all the work be redone, greatly reducing the references. Illustrations of the phenomena would certainly make the reading lighter and more comprehensive. Put more recent references.

Remembering that the work must be understood both by the researchers and also by those who only do clinical work, as the subject guides the decisions to be taken.

As the work will be published in a journal, the number of pages must be condensed, as well as the references.

  Perhaps a systematic review would become the most interesting work, as long as the question was more objective.

Anyway, I believe that the work can be redone to reach the quality it deserves for publication in Biomolecules.

Reviewer 3 Report

In this narrative review, the authors elucidated the effects of initiating neurogenic inflammation (NI) on apical periodontitis (AP) and provided knowledge of neuro-immune crosstalk in chronic AP. Furthermore, they also review the important biomolecules associated with pain signals, immune-neuro interactions (nociceptors), inflammation and immune cells. This review is clear, interesting, enlightening, and worthy of consideration. A minor revision suggestion is listed below:

Since IL-6 receptors are expressed in macrophages. Did you consider adding the content about the effects of ciliary neurotrophic factor (CNTF) on macrophages and apical tissue such as cementoblasts? CNTF is a cytokine of the interleukin-6 family. 

Round 2

Reviewer 2 Report

I would like to thank you for trying to collaborate for a better quality of the paper. I noticed that there was an improvement, but in my point of view, it needs more improvement.

Title

The authors kept the same title stating that the focus of the review is on Nociceptor-Macrophage Interactions in Apical Periodontitis, however the review digresses on several subjects that escape this main focus. Therefore, I still think that the title should be more impactful, since that way it does not become attractive to the reader.

Abstract

I am sorry to inform you but the purpose in abstract is still not clear, and no changes were seen in the revised version of the manuscript.

Lines 70-71 – (ref 18,19) could be taken out of the text in order to shorten the text without losing meaning.

Lines 78 – 80 – The authors are repetitive about the cell influx during the inflammatory process, this keeps the text large, making it tiring to read and not being that attractive.

Lines 87 – 90 – The authors inserted this phrase out of the context of macrophage differentiation and do not add to the understanding of the text, it could be removed.

Lines 90 – 91 to author 28 – Again talking about inflammatory cells, repetitive and could be summarized.

Line 98 – What is PAF, abbreviations must be fully described the first time they appear in the text.

Line 105-109 – Could be rewritten or dropped.

Lines 134 – 138 – The authors assumed that the AP is dynamic in lines 68-75, why then be repeated at this point? Leaving the text exhaustively large, then that paragraph can be removed.

Item 2 of the review (overview) should be more concise, highlighting only the nociceptor cells and macrophages that are the focus of the review, thus reducing extremely confusing parts that may confuse the reader.

The authors made efforts to improve the manuscript. However, even in its present form, it is exhaustively large, containing a lot of information that is not necessary for understanding the focus. I remind you that the paper aims to provide a comprehensive understanding of the temporal role of nociceptors and different clinical manifestations in chronic AP. (lines 54-55) Please reduce the number of references and the number of pages in the paper.

The conclusion of an article should be enough in a paragraph with few sentences, concluding only what was described in your work.
